# Synthetic Route to Conjugated Donor–Acceptor Polymer Brushes via Alternating Copolymerization of Bifunctional Monomers

**DOI:** 10.3390/polym14132735

**Published:** 2022-07-04

**Authors:** Anna Grobelny, Karolina Lorenc, Łucja Skowron, Szczepan Zapotoczny

**Affiliations:** Faculty of Chemistry, Jagiellonian University, Gronostajowa 2, 30-387 Kraków, Poland; grobelnyania@gmail.com (A.G.); karolina.lorenc98@gmail.com (K.L.); l.skowron16@gmail.com (Ł.S.)

**Keywords:** donor–acceptor polymers, polymer brushes, alternating copolymers, RAFT polymerization, metal-free ATRP polymerization

## Abstract

Alternating donor–acceptor conjugated polymers, widely investigated due to their applications in organic photovoltaics, are obtained mainly by cross-coupling reactions. Such a synthetic route exhibits limited efficiency and requires using, for example, toxic palladium catalysts. Furthermore, the coating process demands solubility of the macromolecules, provided by the introduction of alkyl side chains, which have an impact on the properties of the final material. Here, we present the synthetic route to ladder-like donor–acceptor polymer brushes using alternating copolymerization of modified styrene and maleic anhydride monomers, ensuring proper arrangement of the pendant donor and acceptor groups along the polymer chains grafted from a surface. As a proof of concept, macromolecules with pendant thiophene and benzothiadiazole groups were grafted by means of RAFT and metal-free ATRP polymerizations. Densely packed brushes with a thickness up to 200 nm were obtained in a single polymerization process, without the necessity of using metal-based catalysts or bulky substituents of the monomers. Oxidative polymerization using FeCl_3_ was then applied to form the conjugated chains in a double-stranded (ladder-like) architecture.

## 1. Introduction

The development of conjugated polymers (CP) is of particular importance to the solution of crucial worldwide challenges as they have found application in many various devices, such as solar cells [1], light-emitting diodes [2], energy storage [3], sensors [4], wearable equipment [5], and implantable devices [6], which can address health-, environment-, or energy-related issues. Such a wide spread of application fields is determined mainly by advantages of conjugated macromolecules, which include primarily low costs [7], flexibility [5], and facile processability [8]. One particular type of CP are copolymers built from alternating electron-donating and electron-withdrawing monomers, which were introduced by the Havinga group and are called donor–acceptor (D–A) polymers [9]. Apart from all desired properties assigned to their analogous homopolymer, they exhibit some extraordinary features, especially low energy bandgap [10]. The reduction of gap width could be achieved due to the interaction between the donor and acceptor molecules, which results in the hybridization of the HOMO and LUMO orbitals of both moieties and the formation of chains with a stabilized quinonoid form [11]. Furthermore, macromolecules composed of alternating donor and acceptor moieties reveal improved optical and electronic behavior. They are characterized by high molar extinction coefficient [12] and strong absorption of long wavelength of visible near-infrared light [13], as well as enhanced mobility of charges [14] with intramolecular charge transfer, due to strong push–pull effect [15]. That is why they are considered one of the most promising materials for the construction of efficient light-harvesting systems [16].

The proper adjustment between energy levels in D–A polymers is possible, thanks to a variety of monomers available for polymerization. Thiophene-based moieties, such as fused-ring benzodithiophene or cyclopentadithiophene, are commonly used electron-rich moieties [17]. They owe a donor character to a sulfur lone pair, which contributes to the *π* system, and they are commonly applied as macromolecules based on thiophene, which are characterized by a dominating quinoidal character [17] and desired edge-on orientation [18]. Acceptor molecules are often based on the thiadiazole heterocycle, for example, difluorobenzothiadiazole [17]. The combination of these two types of moieties along the chain ensures a noncovalent intramolecular interaction between sulfur and nitrogen atoms, which have an impact on polymer assembly [19]. What is more, the features of both moieties could be further adjusted by proper selection of substituents [20] and are successfully applied in low-bandgap donor–acceptor polymer formation [10].

The methodology applied to the synthesis of D–A polymers should provide a possibility of alternating arrangement of monomers along the chain. That is why the number of available techniques is rather limited to catalyzed cross-coupling reactions [21] conducted according to Stille [22], Suzuki [23], and Sonogashira [24] or direct arylation [25] mechanisms. They all demand rather harsh conditions, giving final products limited efficiency, sometimes in a tedious multistep procedure. Moreover, such kind of synthesis requires the usage of expensive and environmentally harmful palladium or other metal-based catalysts, whose residuals could affect the parameters of the final material that are crucial in their electronic [26] and biomedical applications [27]. Importantly, the monomers need to be substituted by, for example, long alkyl chains to ensure solubility, providing the processability of the obtained polymers [28]. The polymer solubility in an organic solvent is crucial for their applications, as the design of most devices utilize conductive polymer layers deposited from solutions [29].

The settling process is realized mainly using relatively easy, cheap, and fast coating techniques [29]. Nevertheless, organic films obtained using such techniques are not covalently bound to the surface; hence, they have limited stability and may be prone to dewetting [30]. Moreover, they consist of tangled polymer chains that disturb the energy and/or charge transfer process [31], and a precise arrangement of the active moieties within such films would mimic the natural light-harvesting systems that are highly desired [32]. The electronic properties of casted polymer layers are also obstructed by the presence of side bulky substituents, introduced to provide the solubility of the macromolecules [33]. These undisputed problems could be overcome by the formation of conjugated surface-grafted polymer brushes (PB), whose architecture takes into account the presence of molecular wires connected to the substrate via chemical bonds in a perpendicular orientation with respect to the surface. This type of arrangement of polymer chains was proved to ensure much better stability [34] and electrical performance [35], when compared with similar macromolecules obtained in solutions.

PB could be obtained using two general approaches, namely, the “grafting to” and “grafting from”. While the former method requires synthesis of soluble macromolecules, which are next tethered to a surface, the latter one is based on the growth of polymer chains directly from the substrate decorated with appropriate initiator molecules [36]. Using the “grafting from” technique creates an opportunity to achieve a higher grafting density of macromolecules, as there is no steric hindrance, which can impede the process of deposition of new chains on the substrate [36]. However, the characterization of chains connected to the surface is more challenging than those dissolved previously in a solvent [37]. The surface-initiated polymerization approach is usually used to obtain PB, applying, among others, ATRP (atom transfer radical polymerization) [38], RAFT (reversible addition–fragmentation chain transfer polymerization) [39], NMP (nitroxide-mediated polymerization) [40], or PIMP (photoiniferter-mediated polymerization) [35] techniques.

The synthesis of conjugated PB is, however, more challenging and demands the application of other synthetic paths, as radical polymerizations typically cannot lead to the direct formation of conjugated macromolecules [41]. That is why the growth of the chains is realized mainly by Kumada catalyst transfer polycondensation or oxidative polymerization [42]. Nevertheless, the application of a so-called self-templating approach allowed us to use RDRP (reversible-deactivation radical polymerization) techniques to obtain polymer brushes with pendant groups, which were then subjected to reactions to create conjugated chains attached to nonconjugated ones. Such procedure was successfully applied to the syntheses of PB with thiophene-based monomers linked by oxidative polymerization with FeCl_3_ [35,38] or acetylene ones, where a conjugated chain was generated by means of rhodium-catalyzed polymerization [34]. Moreover, macromolecules obtained this way form a ladder-like architecture, which is characterized by an even more ordered structure and higher stability [43].

The methodology of the synthesis of alternating D–A polymer brushes not only should fulfill all requirements imposed to get conjugated chains, but also must ensure an alternating arrangement of monomers along the chain. We reported already the methodology of the synthesis of D–A PB using the Sonogashira, Stille, and Click stepwise processes [44]. The proposed methodology does not require the introduction of any bulky substituent, as it is based on the “grafting from” approach, which does not require solubility of the formed chains. Nevertheless, the whole process is rather tedious and time-consuming, leading to brushes with limited thicknesses, entailing the application of heavy metal catalysts.

Here, we present a proof of concept of the synthetic methodology leading to alternating D–A PB by using surface-initiated reversible-deactivation radical polymerization technique of the alternatingly polymerizable monomers. Styrene and maleic anhydride derivatives were used, as they are known for their tendency toward alternating copolymerization [45]. Thiophene (electron donor) and benzothiadiazole (electron acceptor) were attached to styrene and maleic anhydride, respectively, and the copolymer brushes were formed using surface-initiated RAFT polymerization and metal-free ATRP. The influence of the reaction time and grafting density on the thickness of the polymer layer was investigated using spectroscopic ellipsometry and atomic force microscopy, while the elemental composition of the material was proved by XPS and IR measurements. Finally, the pendant donor and acceptor groups were linked by means of oxidative polymerization with FeCl_3_. The methodology enables the formation of PB with a broad range of thicknesses due to the chain-growth mechanism of the applied polymerization. Moreover, the formation of conjugated grafted chains with an alternating sequence of mers is very facile and does not need the application of metal-based catalysts.

## 2. Materials and Methods

### 2.1. Materials

10-Phenylphenothiazine (PTH, ≥98%), 2-bromo-2-methylpropionyl bromide (BIB, 98%), 2-(dodecylthiocarbonothioylthio)-2-methylpropionic acid *N*-hydroxysuccinimide ester (98%), (3-aminopropyl)triethoxysilane (APTES, 99%), (3-chloropropyl)triethoxysilane (ClPTES, 95%), acetic anhydride (Ac_2_O, 99%), AIBN (0.2 M in toluene), chloroform-*d* (CDCl_3_, 99.8 atom % D), dimethyl sulfoxide-*d* (DMSO-*d*_6_, 99.9 atom % D), silica gel (high-purity grade, average pore size: 60 Å (52–73 Å), 70–230 mesh (63–200 μm, for column chromatography), sodium acetate (CH_3_COONa, 99%), sodium sulfate (Na_2_SO_4_, anhydrous, granular, ≥99.0%), sulfuric acid (H_2_SO_4_, 95–97%), and triethylamine (TEA, ≥99.5%) were all purchased from and delivered by Sigma-Aldrich (St. Louis, MO, USA). Tetrahydrofuran (THF, 99.5%, extra dry over molecular sieves, stabilized) was purchased from Acros Organics (Geel, Belgium). 3-Bromothiophene (97%), 4-ethenylphenylboronic acid (97%), 5-aminobenzo[c][1,2,5]thiadiazol (95%), maleic anhydride (99%), *N*,*N*-dimethylacetamide (DMAC, anhydrous, 99%), tetrakis(triphenylphosphine)palladium(0) (Pd(PPh_3_)_4_, 98%) were all obtained from Fluorochem Ltd. (Hadfield, UK). Ammonia (NH_3_, 30%, cz.d.a.), chloroform (CHCl_3_, p.a.), dichloromethane (DCM, p.a.), diethyl ether (Et_2_O, p.a.), ethanol (EtOH, p.a.), ethyl acetate (p.a.), hexane (p.a.), hydrogen peroxide (H_2_O_2_, 30%), methanol (MeOH, p.a.), *N*,*N*-dimethylformamide (DMF, p.a.), sodium carbonate (Na_2_CO_3,_ p.a.), toluene (p.a.), and tetrahydrofuran (THF, p.a.) were ordered from Chempur (Piekary Slaskie, Poland).

### 2.2. Methods

An Avance III HD (400 MHz) spectrometer (Bruker, Santa Barbara, CA, USA) was applied to register NMR spectra. All data are presented in reference to the solvent residual signal of chloroform-*d* (δ = 7.26 ppm in case of ^1^H NMR and δ = 77.16 ppm for ^13^C NMR) or DMSO-*d*_6_ (δ = 2.50 ppm in case of ^1^H NMR and δ = 39.52 ppm for ^13^C NMR). FTIR spectra were recorded using a Nicolet iS10 spectrometer with an MCT/A detector (Thermo Fisher Scientific^TM^, Waltham, MA, USA). In the case of monomer samples, the ATR accessory (Smart iTX) was applied (number of scans: 128, range: 650–4000 cm^−1^, resolution: 4 cm^−1^, gain: 2.0), while for polymer brushes, the grazing-angle reflectance accessory was used (number of scans: 128, range: 650–4000 cm^−1^, resolution: 8 cm^−1^, gain: 2.0). All spectra were presented after the baseline correction performed using the Omnic software. A MicrOTOF II (Bruker, Bremen, Germany) mass spectrometer with atmospheric pressure chemical ionization (APCI) and a time-of-flight analyzer as a detector were used to record high-resolution mass spectra of synthesized monomers. The absorbances of the monomer solutions (transmission mode, range: 200–800 nm, data interval: 1 nm, scan rate: 600 nm/min) and polymer brushes prepared on a quartz substrate (transmission mode, range: 200–1100 nm, data interval: 1 nm, scan rate: 150 nm/min) were measured by a Varian Cary 50 UV–VIS spectrometer (Agilent Technologies, Santa Clara, CA, USA). A spectroscopic ellipsometer M200U (J. A. Woollam, Lincoln, NE, USA) was applied to evaluate the thickness of the polymer brushes on silica surface. All values are presented as arithmetic means of at least 5 measurements fitted to the Cauchy model in the range 550–1000 nm in the CompleteEASE software. All AFM pictures were collected using a Dimension Icon atomic force microscope (Bruker, Santa Barbara, CA, USA) working in a soft PeakForce Tapping^®^ mode with a ScanAsyst-Air cantilever (nominal force constant: 0.4 N/m). Presented thickness data are arithmetic means of at least 10 measurements, evaluated using the NanoScope Analysis software (Bruker, Santa Barbara, CA, USA). To register XPS spectra, a PHI 5000 VersaProbe II spectrometer (ULVAC-PHI; Chigasaki, Japan) was applied with the monochromatic AlK_α_ radiation source (E = 1486.6 eV). The presented values are calculated as arithmetic means of three measurements. Surftens Universal equipment (OEG GmbH, Frankfurt, Germany) were used to estimate the contact angle measurements. Presented results were calculated as an arithmetic mean of at least 30 measurements, performed in three locations. For the photopolymerizations, the set of LEDs emitting at λ_max_ = 400 ± 5 nm was used, and the light intensity at the samples’ surface equal to 18 ± 2 W/m^2^ was measured by a Delta OHM HD2302.0 light meter equipped with a probe sensitive for the spectral range 400–1050 nm.

### 2.3. Procedures

#### 2.3.1. Synthesis of Monomers

All reactions were conducted under argon atmosphere. Compounds signed as ^N^, to the best of our knowledge, have not been reported in the literature so far (based on SciFinder^®^ and Reaxys^®^ database).

##### 3-(4-Ethenylphenyl)thiophene (**St-D**)

Compound **St-D** was synthesized according to the modified procedure reported in the literature [46]. A solution of 4-ethenylphenylboronic acid (1.5 g, 10.1 mmol) in EtOH (5 mL) was added to the mixture of 3-bromothiophene (1.5 g, 9.2 mmol), 2M Na_2_CO_3_ (9.2 mL, 18.4 mmol), and Pd(PPh_3_)_4_ (0.32 g, 0.3 mmol) in DMF (18 mL). The reaction was left to proceed overnight at 80 °C. After cooling down to the RT, it was neutralized with 30% H_2_O_2_ (5 mL), diluted with water, and extracted with Et_2_O (3 × 150 mL). Combined organic layers was dried over anhydrous Na_2_SO_4_ and concentrated in vacuo. The obtained crude material was purified by column chromatography (silica gel, hexane) to get final product 1 as white solid (1.5 g, 85%). ^1^H NMR (600 MHz, chloroform-*d*) δ 7.57 (m, 2H), 7.45 (m, 3H), 7.40 (m, 2H), 6.74 (dd, *J* = 17.6, 10.8 Hz, 1H), 5.78 (dd, *J* = 17.6, 0.9 Hz, 1H), 5.26 (dd, *J* = 10.8, 0.9 Hz, 1H). ^13^C NMR (151 MHz, chloroform-*d*) δ 142.1, 136.5, 135.4, 128.9, 127.4, 127.1, 126.8, 126.6, 126.4, 126.3, 120.3, 113.8. HRMS (APCI^+^): [M+H]^+^ calculated for C_12_H_11_S^+^: 187.0576; found: 187.0551. HRMS (APCI^−^): no ionization.

##### N-(benzo[c][1,2,5]thiadiazol-5-yl)pyrrole-2,5-dione (**Ma-A**)^N^

To the mixture of 5-aminobenzo[c][1,2,5]thiadiazol (3.1 g, 20.4 mmol) in anhydrous THF (2 mL), the solution of maleic anhydride (2.0 g, 20.4 mmol) in anhydrous THF (25 mL) was added dropwise. The reaction mixture was stirred overnight at RT, cooled down to 5 °C, and left for crystallization. The obtained intermediate was filtered and next diluted in acetic anhydride (6.7 mL, 3.4 mmol). The sodium acetate (0.7 g, 8.2 mmol) was added to the prepared solution, and the reaction was left to proceed overnight at 70 °C and cooled down to 5 °C. The precipitated yellowish product was filtered, washed several times with water, and dried in vacuo (3.5 g, 73%). ^1^H NMR (600 MHz, DMSO-*d*_6_) δ 8.20 (dd, *J* = 9.3, 0.7 Hz, 1H), 8.10 (dd, *J* = 2.0, 0.7 Hz, 1H), 7.74 (dd, *J* = 9.3, 2.0 Hz, 1H), 7.29 (s, 2H). ^13^C NMR (151 MHz, DMSO-*d*_6_) δ 169.6, 153.8, 152.9, 135.0, 133.0, 128.8, 121.2, 118.3. HRMS (APCI^+^): [M+H]^+^ calculated for C_10_H_6_N_3_O_2_S^+^: 232.0175.; found: 232.0169. HRMS (APCI^−^): [M]^−^ calculated for C_10_H_6_N_3_O_2_S^−^: 231.0102; found: 231.0104.

#### 2.3.2. Substrate Preparation

The substrates used for grafting polymer brushes were cleaned according to the protocol described before by our team [44]. Briefly, all samples were sonicated for 10 min in EtOH. After drying under the stream of argon, the silica and quartz substrates were inserted into the so-called piranha solution (30% H_2_O_2_ and H_2_SO_4_|*v*:*v*, 1:3) for 15 min at RT, while ITO samples were put into a heated 50 °C mixture of 30% H_2_O_2_ and 30% aq. NH_3_ (*v:v*, 1:1) for 1 h. Then, the substrates were washed out by a copious amount of water, THF, and toluene. Dried samples were then dealt with the source of air plasma (so-called plasma pen) for 1 min, followed by 30 min UV–ozone cleaner treatment.

#### 2.3.3. APTES Deposition for the Formation of the Initiator Monolayer

The APTES monolayer was deposited using the procedure described before by our team [44]. APTES (50 μL, 0.2 mmol) was added dropwise to a vessel containing substrates immersed in toluene (10 mL). The reaction mixture was left for 2 h at RT, and modified plates were sonicated in toluene for 10 min, washed carefully with DCM, and initially dried under argon stream. Then, they were placed in vacuo for 24 h.

#### 2.3.4. Initiator Monolayer for Surface-Initiated RAFT Polymerization

The substrates modified with the APTES monolayer were immersed in a solution of TEA (10 μL, 0.07 mmol) in ethyl acetate (6 mL). The 2-(dodecylthiocarbonothioylthio)-2-methylpropionic acid *N*-hydroxysuccinimide ester (23 mg, 0.05 mmol) was dissolved in ethyl acetate (4 mL), added dropwise to the reaction mixture, and heated to 50 °C. After proceeding overnight, all samples were sonicated for 5 min in ethyl acetate, rinsed with a copious amount of MeOH, and dried under the stream of argon.

#### 2.3.5. Surface-Initiated RAFT Polymerization

The substrate decorated with the initiator monolayer was placed into a heated pressure tube equipped with a stirring bar. Then, the monomers **St-D** (18.6 mg, 0.1 mmol) and **Ma-A** (23.1 mg, 0.1 mmol) were added. The reactants were next dissolved in a mixture of dioxane and water (2 mL, *v*:*v*, 2:1), and 0.2 M AIBN in toluene (5.0 μL, 0.001 mmol) was added. The polymerization proceeded for a given time at 60 °C, and the sample was cleaned by sonication in THF and toluene and, finally, dried under the argon stream.

#### 2.3.6. Initiator Monolayer for Surface-Initiated Metal-Free ATRP Polymerization

The formation of the initiator monolayer for ATRP was described previously by our team [38]. After APTES deposition, the plates were inserted into a solution of TEA (0.4 mL, 3.0 mmol) in anhydrous DCM (10 mL). Then the mixture of BIB (0.37 mL, 3.0 mmol) in anhydrous DCM (2 mL) was added dropwise. After 1 h, the substrates were cleaned by 5 min sonication in DCM, rinsed carefully with MeOH, and dried under the stream of argon.

#### 2.3.7. Surface-Initiated Metal-Free ATRP Polymerization

The monomers **St-D** (29.8 mg, 0.16 mmol) and **Ma-A** (37.0 mg, 0.16 mmol) were dissolved in 0.3 mL anhydrous DMAC and added to PTH (3.0 mg, 0.01 mmol). Then, one drop of the prepared reaction mixture was deposited on each substrate modified with the initiator monolayer, and the microscopic cover glass was quickly placed on top of the sample, which was next irradiated with a LED source (λ_max_ = 400 ± 5 nm) for a given time. All procedures were performed without access to ambient light. After completing the polymerization process, the plates were sonicated for 5 min in THF and toluene and dried under the argon stream.

#### 2.3.8. Oxidative Polymerization

The samples of the formed grafted alternating copolymer brushes were immersed in the solution of FeCl_3_ (20 mg, 0.12 mmol) in freshly distilled CHCl_3_ (10 mL) to realize oxidative polymerization of the pendant groups. In the case of the samples obtained via RAFT, the mixture was stirred for 1 h at 0 °C; then it was left to proceed overnight at 5 °C and in the next 24 h at RT. The brushes prepared using metal-free ATRP underwent the reaction at 40 °C for 48 h. The obtained samples were taken out in the glove box and rinsed with a copious amount of CHCl_3_ and MeOH and dried under the stream of argon. All the steps were realized in dimmed light.

#### 2.3.9. Change of Grafting Density

To change the grafting density of the polymer brushes, they were grafted from the mixed monolayers containing the initiator, as described in our previous work [44]. To form the mixed monolayer, APTES (50 μL, 0.21 mmol) and ClPTES (50 μL, 0.21 mmol) were dissolved in toluene (10 mL). Next, the solutions were added to the reaction flask in various ratios (*v:v*|3:1, 1:1, 1:3), and the deposition of the monolayers was conducted following the procedure described above.

## 3. Results and Discussion

### 3.1. Synthesis of Donor and Acceptor Monomers

In order to obtain a ladder-like alternating D–A PB, proper monomers containing polymerizable groups and an electron-rich or electron-poor substituent were designed. First, to ensure alternating arrangement of mers styrene (St) and maleic anhydride (MA), molecules were chosen as cores of the monomers for further derivatization with donor and acceptor moieties, respectively. Due to a much higher rate constant of copolymerization than the respective rate constants of homopolymerization processes, St and MA tend to form alternating copolymers [45]. Because of its electron-donating character, St was coupled with the selected donor side group, thiophene, by means of Suzuki reaction giving 3-(4-ethenylphenyl)thiophene (**St-D**). Benzo[c][1,2,5]thiadiazol, selected as an acceptor unit, was attached to MA in a two-step process, consisting of amidation with an amine derivative, followed by ring closure in the presence of acetic anhydride to obtain the final *N*-(benzo[c][1,2,5]thiadiazol-5-yl)pyrrole-2,5-dione (**Ma-A**) (see Figure 1). The structure and purity of both synthesized monomers were confirmed by recorded NMR and IR spectra (see Appendix A).

### 3.2. Synthesis of D–A Polymer Brushes via Surface-Initiated RAFT Polymerization

The preparation of a proper initiator monolayer, necessary to obtain polymer brushes via the “grafting from” approach, was realized by deposition of the APTES monolayer, according to the procedure reported previously by our team [44], followed by attachment of the chain transfer agent (CTA), 2-(dodecylthiocarbonothioylthio)-2-methylpropionic acid *N*-hydroxysuccinimide ester, via the amide bond formation (see Figure 2). The selection of trithiocarbonate CTA was dictated by the character of the synthesized monomers, which can be treated as activated monomers, as the C=C polymerizable groups are conjugated with aromatic ring or carbonyl groups [47]. The surface reactions leading to the formation of the initiator monolayer were followed by the changes in the monolayer thickness and contact angle measurements, which both indicate successful attachment of CTA (see Table 1).

Then, the substrates modified with the initiator monolayer were used to obtain PB using surface-initiated RAFT polymerization. According to the spectroscopic ellipsometry measurements, the layer with a thickness equal to 34.4 ± 0.6 nm was synthesized in 4 h of polymerization at 60 °C in a dioxane/water mixture (*v:v*|2:1). The total concentration of both monomers was equal to 0.1 M, and AIBN was added to the solution as a free initiator (molar ratio n_St-D_: n_Ma-A_: n_AIBN_|100:100:1). The thickness of the obtained polymer chains was confirmed by atomic force microscopy (AFM) technique, which revealed 40.8 ± 0.5 nm. A small increase in the contact angle of the formed brushes with respect to the initiator layer indicates some increase in the hydrophobicity due to the brush formation (see Table 1). The finally applied conditions were optimized in a multifactor process, including parameters such as solvent composition, concentration of all species, type of free initiator, and temperature. Consequently, the addition of water to the reaction system appeared to be crucial for efficient growth of polymer brushes while keeping the maximum concentration of the monomers in the given solvent mixture.

The chemical composition of the obtained coatings was characterized by IR spectroscopy and XPS analysis. The first technique could not lead to the conclusion about the presence of both applied monomers in the polymer chains, as bands typical for the functional groups of both monomers (thiophene, benzo[c][1,2,5]thiadiazole, or styrene) are usually observed in the same wavenumber region (see Appendix A). More information on the actual composition of the chains can be derived from XPS results, which are generally in agreement with the simulated results for the alternating poly(**St-D**-*alt*-**Ma-A**) brushes (equal number of donor and acceptor mers). It is particularly clear when the nitrogen-to-sulfur ratio is considered. It is supposed to be 3:2 = 1.5 (one sulfur atom is present in **St-D** structure, while **Ma-A** contains three nitrogen and one sulfur atoms), while the very close value, 1.61, was found (see Table 2). Moreover, sulfur bonded with carbon and those connected with nitrogen are observed in the same quantity, which also indicates the formation of the aimed alternating copolymer (see Table 2 and Appendix A).

In order to check the correlation between the reaction time and the thickness of the obtained poly(**St-D**-*alt*-**Ma-A**) PB, the set of polymerizations of various times was performed. All prepared samples were then characterized by spectroscopic ellipsometry and AFM to estimate their thicknesses (see Appendix A for exemplary ellipsometric data). Both applied methods revealed the expected growth of chain length with increasing polymerization time (see Table 3 and Figure 1). Moreover, the relationship between the thickness and polymerization time was found to be linear up to 12 h, indicating a controlled character of the process (see Figure 2). The deviations from linearity for longer polymerization times are quite commonly observed and could be caused by increasing steric hindrance, which impedes monomer access to growing chains, as well as by limited mobility of grafted macromolecules [48].

### 3.3. Changing of Grafting Density

The influence of grafting density on the growth of polymer brushes was also investigated. The distance between the grafted chains was changed by using a mixed monolayer approach, applied successfully by our team before [44]. Briefly, during the initiator deposition, the APTES solution was mixed with analogous ClPTES molecules having a terminal chlorine atom. An introduced halogenated derivative is not active in reaction with CTA, contributing to the generation of a lower density of attached CTA groups on the surface. Properly modified substrates with different concentrations of initiating sites on the surface were then used to perform RAFT polymerizations. All samples were prepared in the same optimized conditions during 4 h of polymerization. As a result, the thickest organic layer was obtained in the case of 80% grafting density, which revealed the thicknesses higher by almost 20% than for the substrate with the smallest distance between polymer chains (100% grafting density) (see Table 4 and Appendix A). Such observation can be explained by the reduced steric hindrance during polymerization for the less-crowded growing brushes and lower chance of recombination of the neighboring chains that may significantly enhance the termination process for a high surface concentration of the initiating sites [49]. Nevertheless, further surface dilution of the initiating sites (70% grafting density) caused a significant drop of polymer thickness, likely due to a collapse of the brushes adopting mushroom or even pancake conformations [37]. Changing the grafting density has also an impact on the roughness of the obtained coatings, which, according to the AFM measurements, rises by increasing the distance between polymer chains. The reason for the observed trend could be related to the larger variations of the layer thickness due to the presence of ungrafted spaces at ClPTES-modified sites.

### 3.4. Synthesis of D–A Polymer Brushes via Metal-Free ATRP

The proposed approach based on the application of styrene and maleic anhydride as a platform for the formation of donor–acceptor polymer brushes was realized also by means of a metal-free ATRP technique. This methodology was introduced by Hawker et al. [50], and it allows for covering a wider area with grafted macromolecules using only microliter volumes of reaction mixtures, while in order to perform the RAFT process, significantly higher amounts of the synthesized monomers are required. First, the substrates were coated with the initiator monolayer composed of APTES and terminal BIB groups. Then, polymerization was conducted in the solution of **St-D** and **Ma-A** monomers (c = 0.5 M of each monomer) and a PTH agent in the ratio n_St-D_: n_Ma-A_: n_PTH_|30: 30: 1. The alternating copolymer brushes with a thickness of 13.5 ± 1.2 nm were prepared in 4 h of the polymerization. The presence of an organic layer with 13.4 ± 0.2 nm thickness and low roughness (R_q_ = 4.7 nm, R_a_ = 3.7 nm) was also confirmed by the captured AFM images (see Figure 3), and the collected IR spectrum overlaid the one for the brushes synthesized via RAFT (see Appendix A). Furthermore, the revealed linear character of correlation between the reaction time and thickness of the polymer layer indicates a controlled nature of the whole process (see Appendix A). Presented results prove the versatility of the proposed approach of the synthesis of donor–acceptor polymer brushes using maleic anhydride and styrene derivatives; however, a metal-free ATRP technique is characterized by smaller monomer consumption, slower rate, and higher smoothness of the obtained PB.

### 3.5. Oxidative Self-Template Polymerization of Polymer Brushes

Synthesized polymer brushes with pendant donor and acceptor groups were next used as multimonomeric macromolecules in a template polymerization to obtain ladderlike PB. In order to generate a linkage between side thiophene and benzo[c][1,2,5]thiadiazol rings, FeCl_3_ as an oxidative agent was applied; as such, an approach was successfully implemented to synthesize polythiophene chains [35,38,42]. The observed growth of the layer thickness from 34.4 ± 0.6 to 41.2 ± 2.0 and increase in the water contact angle after conjugation of poly(**St-D**-*alt*-**Ma-A**) brushes obtained in the RAFT process indicate a successful polymerization (see Table 1). Such an effect could be assigned to stretching and stiffening of polymer chains due to the adaptation of a double-stranded architecture, as it was observed previously by our team in the case of other brushes obtained in a self-templating approach [35]. Moreover, collected AFM images for the samples with reduced grafting density revealed substantial changes in the morphology after FeCl_3_ treatment as distinguishable bridges between polymer domains (see Figure 4). Their presence could be related to the contribution of intermolecular connections formed during the oxidative polymerization. The formation of such connections between such stretched chains also seemed to improve the uniformity of the layer as the surface roughness decreased substantially (see Figure 4) and the contact angle increased (113°, see Table 1), which can account for a more uniform coating of the substrate by hydrophobic brushes. The oxidative polymerization of the brushes obtained in the ATRP process required more harsh conditions to proceed efficiently (see further for spectroscopic characterization) since the same conditions, as applied for the brushes obtained in the RAFT process, were not sufficient. It seems that poly(**St-D**-*alt*-**Ma-A**) brushes obtained via RAFT polymerization are more susceptible to oxidative polymerization forming double-stranded systems or just bridging the neighboring chains. Their required milder conditions for such a process that could be related to differences in the grafting density between both systems, which determine the steric hindrance and easiness of penetration of the oxidative agent.

### 3.6. Spectroscopic Measurements

The course of the self-templating polymerization was also followed using spectroscopic techniques. The observed increase in the absorbance of the brushes after the oxidative polymerization and the formation of the red-shifted absorption tail could indicate covalent bonding between the pendant groups, leading to conjugated chains for both studied brushes (see Figure 5). Nevertheless, oxidative polymerization did not cause substantial changes (only 1–2 nm) in the position of the band near 310 nm, which could be assigned to the presence of the unreacted acceptor unit (see Appendix A for comparison). Such observation suggests that the thiophene groups have a higher tendency to undergo the polymerization process, while a number of benzothiadiazole rings remain unreacted. It can be explained in terms of the larger resistance of the acceptor groups to oxidation conditions. In the case of polymerization of the ATRP-formed brushes, a higher temperature (40 °C) and longer time (48 h) had to be applied to reach similar spectroscopic changes, as observed for the polymerization of the RAFT-formed brushes. It is likely due to a more packed layer observed for the ATRP-formed brushes (compare Figure 3 and Figure 4), which limits its penetration by the oxidizing agent, which forms a suspension (limited solubility) in chloroform serving as the polymerization medium [51]. While the presented results support the proof-of-concept approach for the facile formation of D–A PB, the polymerization conditions, grafting density, other oxidation agents can be further developed to increase the contribution of the acceptor unit in the formed conjugated chains.

IR spectroscopy also confirmed the formation of the conjugated chains by the appearance of the new weak band near 1575 cm^−1^, after oxidative polymerization, which can be assigned to the stretching vibration of C=C bonds in the conjugated system (see Figure 6). The lack of other noticeable changes in the region typical for the appearance of conjugated chain vibrations (1400–1600 cm^−1^) is caused by the fact that such bands observed for polythiophene brushes have typically low intensity [35], so in the system containing various aromatic rings (phenyl, thienyl, and benzothiadiazole group), they may not be distinguished. The increase in the intensity of bands assigned to C–H stretching vibrations (slightly below 3000 cm^−1^) was noticed after the oxidative polymerization. It can be related to the reorganization of polymer chains into a more extended structure after oxidative polymerization that is in line with the conclusions from the AFM and ellipsometry measurements [35,38,52]. Nevertheless, the lack of a band at ca. 3300 cm^−1^ (hydroxyl groups) indicates no oxidation of the pendant thiophene groups.

## 4. Conclusions

We presented here a new facile methodology of synthesis of donor–acceptor polymer brushes via surface-initiated reversible deactivation radical polymerization techniques. The styrene molecule coupled with a thiophene ring and maleic imide substituted with benzothiadiazole were applied as donor and acceptor monomers, respectively, which formed alternating copolymer brushes in surface-initiated metal-free ATRP and RAFT polymerization. The brushes were grafted from a surface in a single controlled polymerization process, and their thickness could be easily adjusted by varying the polymerization time. The pendant donor and acceptor groups, alternatingly aligned along the stretched polymer chains, were then subjected to oxidative polymerization with an FeCl_3_ agent, leading to the formation of conjugated brushes. The presented proof of concept showed that, this way, the conjugated brushes can be formed as indicated by UV–VIS and IR spectroscopy. However, quantitative introduction of the acceptor, benzothiadiazole, groups into the formed conjugated chain is difficult in the oxidative polymerization process likely due to a higher tendency of thiophene to undergo this process. Nevertheless, the facile methodology is worthy of further development by, for example, varying the polymerization conditions, type of the oxidative agent, or grafting density. The proposed methodology is also very versatile as other donor–acceptor systems can be formed by changing just the pendant donor or acceptor moieties in the monomers, and other polymerization methods (e.g., electrochemical polymerization) can be easily applied to achieve conjugated chains. Importantly, the obtained brushes do not contain even traces of metal atoms, and no bulky substituents are necessary in the formed conjugated brushes, so their properties are not affected by these factors. Such features and versatility of the method, as compared with catalyzed cross-coupling reactions, make it attractive for the synthesis of donor–acceptor thin layers with increasing applicability, especially in the fabrication of various optoelectronic devices.

## Data Availability

All data are contained within the publication or available Appendix A.

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
