# Peer review of "Synthetic Route to Conjugated Donor–Acceptor Polymer Brushes via Alternating Copolymerization of Bifunctional Monomers"

_polymers, 2022, doi:10.3390/polym14132735_

Round 1
Reviewer 1 Report
I read the manuscript provided by the authors and I have to say that I found it very interesting and to the point for polymer science. I have some comments and some questions (see pdf attached).
The major problem was that the supporting information the authors refer to are not uploaded so I could not see them. That is why my recommendation is a major revision until I could actually have a look at the whole manuscript (main and supporting).

Author Response
We would like to thank the Reviewer for the valuable comments and corrections that allowed us to improve the manuscript. We are really confused due to information about the lack of access to the SI file that was submitted with the original manuscript. Please, find below our response to the comments. We have revise the manuscript and the SI file accordingly.
- Lines 33-35: Please add references.
Appropriate reference (nr 10 in the revised manuscript) has been added.
- Lines 46-47: Please add references.
Appropriate reference (17) has been added.
- Lines 84-90: Please add references.
Appropriate reference (36) has been added.
- Line 101: Please define RDRP.
The acronym RDRP has been defined in the main text as requested.
- 3.1: I think all the NMR spectra must be available as supported information.
The NMR spectra have been inserted to the revised SI file.
- 3.2: Any reference for the protocol?
Appropriate citations (references 44 and 38) have been placed in the descriptions of the protocols.
- 3.4-2.3.8: If different works were used for these synthetic approaches,
references must be added.
Appropriate references have been added in the sections: 2.3.2, 2.3.3, 2.3.6, 2.3.9.
- Line 293: The authors refer to supporting information, but no supporting
information uploaded.
We are very sorry, that the supporting information was no available to you. We believe, it was uploaded during the original submission and certainly contained the IR spectra that are referred in line 293. In the revised version of SI file the NMR spectra have been added, and the IR spectra are now in Figure S5.
- Scheme 2: Please adjust the scheme to distinguish better the molecules.
The Scheme 2 has been adjusted (the molecules were bolded and enlarged).
- Line 331: XPS data must be included in supporting information.
The XPS data, as well as deeper insight in the sulfur region, which is crucial for the compositional analysis of the layer, have been added to the revised SI file (Figure S6). The deconvoluted spectrum is presented from which the data in Table 2 were derived.
- Lines 374-376: Please add references.
We have reconsidered the mentioned explanation and added an additional one supporting the statement with an appropriate reference (49). Thus, in addition to possible contribution of the steric hindrance also reduce probability of recombination of the neighboring growing chains may explain the increase of the brush thickness for surface-diluted initiator monolayer as compare to the full initiator monolayer..
- Lines 374-379: I think the authors should also present the AFM images of the
three different grafting densities.
The AFM images of samples with various grafting densities have been added to the SI file (Figure S8) as requested.
- Figure 4: Did the authors measure the contact angle of the precursor and after the oxidative polymerization? I think it would be interesting to show the values. Also, how did the roughness changed after the oxidative polymerization?
Data about changes in contact angle were introduced to the Table 1 and the roughness has been added to Figure 4.
In fact, the observed increase of contact angle (up to 113°) and decrease of surface roughness after oxidative polymerization seem to be the results of formation of the layer that more uniformly coats the substrate. As the brushes are hydrophobic a high water contact angle is expected for a continuous layer formed by such brushes. Appropriate comments have been introduced into the revised manuscript (page 11).
- Lines 426-427: Any reference to support this?
The conclusion was withdrawn from observations of the topography changes after oxidative polymerization as shown in AFM images (Figure 4). We find it reasonable to attribute the presence of the observed wire-like small features to the conformational changes of the neighboring chains related to their covalent bridging. They are distinguishable thanks to the presence of the formed nanodomains of brushes while they are not visible between such domains before oxidative polymerization. Unfortunately, we could not find any similar reports in the literature to further support the claim.
- Lines 449-451: How do the authors know this?
The AFM images (Figure 3 and 4) clearly show the difference in the density of the brushes obtained by ATRP (dense homogenous layer) and RAFT polymerization (nanodomains morphology). As the FeCl3 forms a suspension (limited solubility) in chloroform, denser brushes could limit its diffusion to the lower parts of the material. Appropriate reference has been added to the revised manuscript.
- UV figure is marked as “1”, please correct it.
The label has been corrected.
- Figure UV: Why the peaks at 300 nm are one for the oxidative but two for the RAFT-ATRP.
The maximum near 280 nm, observed for the brushes before oxidative polymerization, correlates with the absorbance of donor St-D monomer (see Figure S11). As a result of the self-templating polymerization isolated thiophene rings tend to disappear forming conjugated chains with red-shifted absorption. The second peak, which can be related to the absorbance of the acceptor Ma-A monomer becomes slightly blue-shifted after the oxidation polymerization but does not disappear.
- Lines 466-468: Please add references.
Similar effect we have observed during our previous investigations. Appropriate references have been added.

Reviewer 2 Report
The manuscript written by Grobelny et al. proposed the synthesis of conjugated donor-acceptor polymer brushes via surface-initiated RAFT polymerization and post oxidative polymerization. But I feel like the conclusions of this work may not be well supported by their data due to the following reasons:
1) The authors claimed the synthesis of “conjugated donor-acceptor polymer brushes”. The polymer brush was actually prepared via SI-RAFT which resulted in just normal vinyl polymer rather than conjugated polymer.
2) Although the authors applied oxidative polymerization after SI-RAFT to crosslink the polymer brushes, the results materials are not polymer brush anymore. In addition, the resultant crosslinked material was not clearly characterized besides rough FTIR spectra. e.g., What’s the crosslinking ratio? How did the morphology change? If a conjugated polymer brush was really obtained, what’s the conductivity?
3) So, I do not know what’s the benefit of “conjugated donor-acceptor polymer brushes” synthesized in this work? Throughout the manuscript, I did not see any data related to the function of the resultant “conjugated polymers” or “donor-acceptor”. In principle, both of them are very interesting and have many applications.
4) Actually, both monomers St-D and Ma-A can be polymerized via SI-RAFT independently. How could the authors obtain a perfect alternative copolymer i.e., poly(St-D-alt-Ma-A) in Scheme 2?
5) In the abstract “The oxidative polymerization using FeCl3 has then applied to the form conjugated chains in a double-stranded (ladder-like) architecture”. How could the author prove the formation of a ladder-like polymer, by just FTIR? What does the structure look like?
6) In summary, the author declared several interesting keywords for their work such as “conjugated polymer brush”, “donor-acceptor polymer brush”, “double-stranded (ladder-like) architecture”, but in their data, none of them was really proven or demonstrated. This manuscript should be largely improved/corrected if the author would like to make a scientific publication.
Round 2
Reviewer 1 Report
I have read the revised manuscript from the authors and I have to say that it has been improved a lot after the incorporation of new references and explaining some points. I think they should recheck the numeration of the figures since they have figures 1-4 and then UV has been marked again as 1 and FTIR as 2.
Author Response
Thank you for positive response on our revision.
As for wrong numbers of some figures it must have happened during transformation from doc to pdf format. We are correcting this problems in the current revision
Reviewer 2 Report
The revised manuscript can be accepted.
Author Response
Thank you very much for a positive response to our revision.